# Role of Hydrogen Sulfide and Polysulfides in the Regulation of Lipolysis in the Adipose Tissue: Possible Implications for the Pathogenesis of Metabolic Syndrome

**DOI:** 10.3390/ijms23031346

**Published:** 2022-01-25

**Authors:** Jerzy Bełtowski, Krzysztof Wiórkowski

**Affiliations:** 1Department of Pathophysiology, Medical University of Lublin, 20-090 Lublin, Poland; 2Luxmed Lublin Medical Center—Primary Care Unit, 20-080 Lublin, Poland; kwiorkowski@gmail.com

**Keywords:** adipose tissue, lipolysis, hydrogen sulfide, polysulfides, obesity

## Abstract

Hydrogen sulfide (H_2_S) and inorganic polysulfides are important signaling molecules; however, little is known about their role in the adipose tissue. We examined the effect of H_2_S and polysulfides on adipose tissue lipolysis. H_2_S and polysulfide production by mesenteric adipose tissue explants in rats was measured. The effect of Na_2_S and Na_2_S_4_, the H_2_S and polysulfide donors, respectively, on lipolysis markers, plasma non-esterified fatty acids (NEFA) and glycerol, was examined. Na_2_S but not Na_2_S_4_ increased plasma NEFA and glycerol in a time- and dose-dependent manner. Na_2_S increased cyclic AMP but not cyclic GMP concentration in the adipose tissue. The effect of Na_2_S on NEFA and glycerol was abolished by the specific inhibitor of protein kinase A, KT5720. The effect of Na_2_S on lipolysis was not abolished by propranolol, suggesting no involvement of β-adrenergic receptors. In addition, Na_2_S had no effect on phosphodiesterase activity in the adipose tissue. Obesity induced by feeding rats a highly palatable diet for 1 month was associated with increased plasma NEFA and glycerol concentrations, as well as greater H_2_S production in the adipose tissue. In conclusion, H_2_S stimulates lipolysis and may contribute to the enhanced lipolysis associated with obesity.

## 1. Introduction

Adipose tissue is one of the most abundant tissues in human body. For a long time considered only as a passive site of energy storage, adipose tissue is now recognized as a very active metabolic and endocrine organ. Fatty acids liberated from triglycerides stored in the adipose tissue represent an important source of energy for many organs. Adipose tissue is one of the target tissues for insulin and, as such, is involved in the regulation of systemic insulin sensitivity and resistance. In addition, adipose tissue produces many hormones referred to as adipokines, such as leptin, resistin, visfatin, adiponectin, etc. [1,2,3,4]. Specific adipose tissue depots also have special functions. For example, perivascular adipose tissue (PVAT), which surrounds the blood vessels, produces vasodilating and anti-inflammatory mediators, which have an important role in maintaining vascular homeostasis; however, obesity is associated with PVAT dysfunction due to its inflammation and local oxidative stress [5,6]. Unlike white adipose tissue, brown adipose tissue (BAT) contains many small lipid droplets and many mitochondria, and is characterized by intensive fatty acid oxidation through uncoupled mitochondrial respiration with little ATP production and a large amount of energy dissipated as heat, making it important in thermogenesis. Interestingly, in addition to fatty acids, BAT can oxidize other metabolites such as glucose, lactate, succinate and branch-chain amino acids [7]. BAT also secretes mediators which improve the metabolism of remote tissues and are referred to as BATokines [8]. Research interest in adipose tissue has expanded due to increasing prevalence of obesity and metabolic syndrome, which is a cluster of obesity-associated abnormalities such as dyslipidemia, impaired glucose tolerance/Type 2 diabetes, arterial hypertension, chronic prothrombotic and pro-inflammatory states, all of which contribute to the development of atherosclerosis, ischemic heart disease, heart failure, nephropathy, neuroinflammation and certain cancers [1,9,10].

Lipolysis is the principal metabolic process in the adipose tissue. In adipocytes, triglycerides are hydrolyzed to glycerol and fatty acids by sequential action of adipose triglyceride lipase (ATGL), hormone-sensitive lipase (HSL) and monoglyceride lipase (MGL). Glycerol is then released to the extracellular space, and fatty acids are either released or re-esterified to triglycerides inside the adipose tissue. Obesity and metabolic syndrome are characterized by enhanced adipose tissue lipolysis and an increased non-esterified fatty acid (NEFA) concentration, which contributes to insulin resistance, dyslipidemia, endothelial dysfunction and lipotoxicity [11,12,13,14]. Lipolysis is a highly regulated process. The main target of regulation is HSL, which is activated through phosphorylation by cAMP-stimulated protein kinase A (PKA). The main activators of this pathway are catecholamines binding to β_2_ and/or β_3_-adrenergic receptors. In addition, HSL may be phosphorylated by cGMP-activated protein kinase G (PKG) in response to factors such as nitric oxide and natriuretic peptides. The main inhibitor of lipolysis is insulin, which activates phosphodiesterase and decreases cAMP concentrations [12,14].

Apart from peptide hormones (adipokines), gasotransmitters, including NO, CO and H_2_S, are also produced in the adipose tissue [15]. The H_2_S produced by perivascular adipose tissue is involved in the regulation of vascular tone [16]. In addition, H_2_S is involved in the regulation of adipogenesis [17], insulin-stimulated glucose uptake [18], adipokine production [19] and adipose tissue inflammation [20]. However, little is known about the role of H_2_S in the regulation of lipolysis.

Recent studies have suggested that, apart from H_2_S, inorganic polysulfides (H_2_S_n_, n = 2–8) are also important signaling molecules. H_2_S_n_ polysulfides are produced by partial oxidation of H_2_S or its interaction with NO, or are directly synthesized enzymatically by 3-mercaptopyruvate sulfurtransferase (MST) [21]. In some experimental systems, polysulfides are much more potent than H_2_S. Indeed, persulfidation of protein cysteine residues, believed to be the main signaling mechanism of H_2_S, can be accounted for by H_2_S_n_ rather than H_2_S itself [21,22,23]. It is unclear if polysulfides are produced in the adipose tissue and, if so, what their function is. In the present study, we examined the role of H_2_S and polysulfides in the regulation of adipose tissue lipolysis.

## 2. Results

### 2.1. H_2_S and Polysulfide Production by Adipose Tissue

H_2_S and polysulfide production was measured by mesenteric adipose tissue explants ex vivo. Very small amounts of H_2_S and H_2_S_n_ were produced by adipose tissue in the absence of L-cysteine (L-Cys) and pyridoxal 5′-phosphate (PLP), which are the substrate and cofactor, respectively, of cystathionine β-synthase (CBS) and cystathionine γ-lyase (CSE). Adipose tissue incubated in the presence of L-Cys and PLP produced more polysulfides than H_2_S (Figure 1). A CSE inhibitor, propargylglycine (PAG), markedly reduced H_2_S but had no effect on H_2_S_n_ production. Adipose tissue incubated in the presence of the MST substrate 3-mercapropyruvate (3-MP), but without L-Cys and PLP, produced less H_2_S than in the presence of L-Cys and PLP. However, H_2_S_n_ production in the presence of 3-MP was similar to that observed in the presence of L-Cys and PLP. Inhibitors of MST, L-aspartate and phenylpyruvate, reduced H_2_S and H_2_S_n_ production in the presence of 3-MP but had no effect on H_2_S production in the presence of L-Cys and PLP. These results suggest that H_2_S is produced in the adipose tissue mainly by CSE and, to a lesser extent, by MST, whereas the latter enzyme is responsible for polysulfide production (Figure 1).

### 2.2. Effect of Na_2_S and Na_2_S_4_ on Non-Esterified Fatty Acids and Glycerol Concentration

Na_2_S injected at a dose of 100 μmol/kg increased plasma NEFA concentrations in a time-dependent manner. The maximal NEFA level was observed 15 min after injection (Figure 2a). In contrast, Na_2_S_4_ had no significant effect on plasma NEFA. Consequently, the area under the curve of plasma NEFA between 0 and 60 min after Na_2_S injection was much higher than in control animals receiving 0.9% NaCl (Figure 2b).

Similarly, Na_2_S but not Na_2_S_4_ increased plasma glycerol concentrations in a time-dependent manner (Figure 3). The effect of Na_2_S_2_ on plasma NEFA and glycerol at 15 min after injection was dose-dependent (Figure 4). The minimal dose of Na_2_S that induced significant increases in plasma NEFA and glycerol was 10 μmol/kg. Taken together, the results suggest that H_2_S, but not polysulfides, stimulates lipolysis in the adipose tissue.

### 2.3. Role of Cyclic Nucleotides and Cyclic Nucleotide-Dependent Protein Kinases in the Regulation of Lipolysis by Na_2_S

Cyclic AMP (cAMP)-dependent protein kinase A (PKA) and cGMP-dependent protein kinase G (PKG) play an important role in the regulation of lipolysis by phosphorylating hormone-sensitive lipase in adipocytes, To examine if cyclic nucleotides and the respective kinases are involved in Na_2_S-stimulated lipolysis, we measured cAMP and cGMP concentrations in the adipose tissue, and examined the effect of protein kinase inhibitors on the lipolytic response to Na_2_S. Na_2_S increased cAMP but not cGMP in adipose tissue (Figure 5a), whereas Na_2_S_4_ had no effect (data not shown). Neither the PKA nor the PKG inhibitor (KT5720 and KT5823, respectively) had a significant effect on plasma NEFA and glycerol if administered to rats not receiving Na_2_S (not shown). However, KT5720 but not KT5823 abolished the Na_2_S-induced increase in plasma NEFA and glycerol (Figure 5b). These data suggest that the cAMP–PKA pathway but not the cGMP–PKG pathway is involved in the stimulation of lipolysis by H_2_S.

### 2.4. Role of the Adrenergic System in the Regulation of Lipolysis by Na_2_S

Norepinephrine released by sympathetic endings, and epinephrine and norepinephrine secreted by the adrenal medulla and circulating in the blood stimulate lipolysis by binding to β-adrenergic receptors and activating the cAMP–PKA pathway. To examine if β-adrenergic receptors are involved in the stimulation of lipolysis by H_2_S, we performed additional experiments using a β-adrenergic agonist, isoproterenol, and a β-adrenergic antagonist, propranolol. Propranolol administered to the rats not receiving Na_2_S had no effect on plasma NEFA and glycerol concentrations (not shown). As expected, isoproterenol administered at 1 mg/kg potently stimulated lipolysis, as evidenced by the marked increase in plasma NEFA and glycerol (Figure 6). The effect of isoproterenol was prevented by earlier administration of propranolol. However, propranolol had no effect on the Na_2_S-induced increase in NEFA and glycerol (Figure 6).

### 2.5. Effect of Na_2_S on cAMP-Hydrolyzing Phosphodiesterase Activity in Adipose Tissue

The intracellular cAMP concentration is dependent not only on its synthesis by adenylyl cyclase but also by its degradation by phosphodiesterases (PDE). Because we have demonstrated that Na_2_S stimulates the cAMP–PKA pathway but β-adrenergic receptors, the main adenylyl cyclase-activating receptors in the adipose tissue, are not involved, we hypothesized that Na_2_S could increase cAMP by inhibiting PDE. However, Na_2_S (100 μmol/kg) had no effect on PDE activity in the adipose tissue (control group: 78 ± 16 pmol/min per mg protein; Na_2_S-treated group: 86 ± 14 pmol/min per mg protein).

### 2.6. Effect of Obesity on Metabolic Parameters, H_2_S and Polysulfide Production in Adipose Tissue

Feeding rats with a high-calorie diet for 1 month resulted in a significant increase in body weight; however, plasma lipids, insulin and glucose concentrations did not differ from those of the control group (Table 1). However, plasma NEFA and glycerol levels were significantly higher in obese rats. The adipose tissue of obese rats produced more H_2_S than that from lean animals. Polysulfide production by adipose tissue explants isolated from obese rats tended to be lower than in lean animals but the difference was not significant (Table 1).

## 3. Discussion

The main findings of this study are that: (1) both H_2_S and polysulfides are produced in the adipose tissue, and, under physiological conditions, the amount of polysulfides is higher than that of H_2_S; (2) H_2_S stimulates lipolysis in a cAMP–PKA-dependent but β-adrenergic receptor-independent manner; (3) obesity induced by a high-calorie diet increases H_2_S but not H_2_S_n_ production in the adipose tissue.

In 2009, it was first demonstrated that H_2_S is produced by various adipose tissue depots, including epidydimal, perirenal and brown adipose tissue in the rat, and that CSE is the main enzymatic source of this gasotransmitter [18]. Since that time, the expression of all H_2_S-synthesizing enzymes (CBS, CSE, MST) and H_2_S production have been demonstrated in various adipose tissue depots in rodents and humans [15]. In addition, Cthe SE-H_2_S system is upregulated during adipogenesis and is involved in this process [24]. We confirmed that H_2_S production in the adipose tissue is mainly accounted for by CSE, as evidenced by the marked inhibitory effect of PAG, as well as much lower H_2_S production in the presence of MST substrate than in the presence of the CSE substrate. On the other hand, MST appears to be the main source of polysulfides. Indeed, H_2_S_n_ production was poorly inhibited by PAG, was observed in the presence of 3-mercaptopyruvate and was markedly inhibited by MST inhibitors. These findings are consistent with recent findings suggesting that MST produces polysulfides or protein persulfides directly but to a much lesser extent H_2_S itself, and that MST is the main source of H_2_S_n_ in the brain [25]. Polysulfides may also be synthesized during partial enzymatic or non-enzymatic oxidation of H_2_S [21]; however, the divergent effects of CSE and MST substrates and inhibitors on H_2_S and H_2_S_n_ suggest that both are produced independently in the adipose tissue and could have different physiological roles.

The role of H_2_S in the regulation of adipose tissue lipolysis has been examined in few studies so far. Geng at al. [26] demonstrated that L-cysteine and the slow-releasing H_2_S donor, GYY4137, inhibited baseline and isoproterenol-stimulated glycerol release by epidydimal adipose tissue in rats and reduced HSL phosphorylation, whereas propargylglycine had the opposite effect. In vivo, propargylglycine increased plasma glycerol concentration in mice fed normal chow, and tended, although not significantly, to increase glycerol in mice fed a high-fat diet, whereas GYY4137 reduced glycerol concentrations only in mice fed a high-fat diet. Ding et al. [27] demonstrated that NaHS reduces basal and isoproterenol-stimulated glycerol release by the epidydimal adipose tissue of mice fed a high-fat diet; however, the effect on adipose tissue collected from lean animals has not been examined. In addition, NaHS reduced HSL phosphorylation, which probably resulted from increased persulfidation of its regulator, perilipinm-1. On the other hand, the adipose tissue of CSE knockout mice exhibited greater glycerol release [27]. These results suggest that H_2_S inhibits lipolysis, which is not consistent with the results of the present study. The reasons for this discrepancy are unclear but several possibilities should be considered. First, H_2_S could regulate lipolysis in a species-dependent manner. Second, our study was performed in lean animals, whereas the study of Ding et al. [27] was in mice fed a high-fat diet. High-fat feeding increases the baseline lipolysis rate, and H_2_S could have different effects on lipolysis in lean and obese animals. Third, the effect of H_2_S on lipolysis could be different in in vitro/ex vivo studies using adipocytes or isolated adipose tissue [27] than in vivo, when endogenous neurohormonal systems regulating lipolysis are active. Finally, we used Na_2_S as the H_2_S donor, whereas Ding et al. [27] used NaHS and GYY4137. According to some studies [28], NaHS is more susceptible to oxidation than Na_2_S, and its solution could contain other more oxidized reactive sulfur species with potentially different effects. Although GYY4137 is considered a more physiologically relevant H_2_S donor, as it releases gasotransmitters slowly and for a prolonged time [29,30], it has also some limitations. In particular, both GYY4137 and the products of its decomposition may have some H_2_S-independent effects [31,32]. In addition, in one study [33], it was demonstrated that GYY4137 does not release H_2_S in rat plasma and, in contrast to Na_2_S, is ineffective in lowering blood pressure in the rat. Moreover, there is no polysulfide donor structurally related to GYY4137; therefore, we used Na_2_S and Na_2_S_4_ as the closely related donors. Interestingly, garlic extract, which contains natural organic H_2_S donors, stimulated lipolysis in 3T3-L1 adipocytes [34]. These data indicate that various H_2_S donors could have different effects on lipolysis.

The results of this study indicate that H_2_S stimulates lipolysis in a cAMP–PKA dependent but β-adrenergic receptor independent manner. The mechanism through which H_2_S increases cAMP in the adipose tissue remains to be established. In adipocytes, adenylyl cyclase is stimulated by the agonists of at least 10 other G-protein-coupled receptors in addition to β-adrenergic receptors [35]. Cai et al. [17] demonstrated that GYY4137 reduced whereas propargylglycine increased phosphodiesterase activity in 3T3-L1 adipocytes. However, we did not observe any effect of Na_2_S on phosphodiesterase activity. Another possibility is direct stimulation of adenylyl cyclase by H_2_S; this effect has been observed in several non-adipose tissue systems [36].

The effect of obesity on the adipose tissue H_2_S system is controversial. Both increases [18] and decreases [26,30,37,38] in the expression of CBS, CSE, MST and adipose tissue H_2_S production have been observed, depending on type of diet, the duration of high-calorie feeding, the animal species and adipose tissue depots. Increased H_2_S excretion in the exhaled air of obese children [39] and increased plasma H_2_S in morbidly obese adults [40] have been reported. Interestingly, in an early study on this topic, Whiteman et al. [41] observed that plasma H_2_S concentration was moderately reduced in overweight or obese patients without diabetes but markedly reduced in obese patients with Type 2 diabetes and insulin resistance. These results suggest that more advanced metabolic syndrome is associated with H_2_S deficiency. Indeed, in studies in which reduced adipose tissue H_2_S has been observed, a high-calorie diet was applied for a long time, leading to severe obesity, dyslipidemia, insulin resistance and/or Type 2 diabetes [26,37]. It should be noted that our model represented relatively mild obesity with no evidence of the metabolic abnormalities characteristic of metabolic syndrome because plasma insulin, glucose and lipid levels were within the normal range. However, plasma NEFA and glycerol concentrations were higher in obese than in control rats, which is consistent with increased adipose tissue lipolysis. Together with higher H_2_S production in the adipose tissue, these findings suggest that H_2_S could be involved in hyperlipolysis in the obese group. However, further studies are required to verify this hypothesis.

One of the important effects of insulin is to suppress lipolysis in the fasting state. Although fasting insulin concentrations were normal in obese rats in our study, suggesting intact insulin sensitivity, we cannot exclude that the antilipolytic effect of insulin in the adipose tissue was specifically impaired. However, this possibility seems unlikely because it is generally appreciated that the effect of insulin on glucose metabolism is impaired earlier during the development of insulin resistance, and that only severe insulin resistance or deficiency impair its effect on lipolysis. Nevertheless, the effect of insulin on adipose tissue should be studied directly in our model to address this issue.

Our study has several limitations. First, the method we used could not discriminate between various polysulfides, and we used only one polysulfide donor (Na_2_S_4_). Studies in other systems have indicated that at least three polysulfides, i.e., H_2_S_2,_ H_2_S_3_ and H_2_S_4_, are produced in tissues [21]. Although various polysulfides could have different effects, this issue has not been widely examined until now. Second, H_2_S and polysulfide production, and cyclic nucleotide concentrations were measured only in one adipose tissue depot. Although various parts of the adipose tissue differ significantly, the mesenteric adipose tissue used by us is a part of the visceral adipose tissue, which is highly lipolytically active and is involved in the pathogenesis of obesity-associated complications [1]. In addition, the effect of PKA and PKG inhibitors were consistent with cyclic nucleotide concentrations, suggesting that the adipose tissue depot used by us was representative regarding systemic regulation of lipolysis by reactive sulfur species. Third, we administered only a single dose of Na_2_S or Na_2_S_4_, whereas obesity is associated with long-term changes in H_2_S/polysulfide levels. However, the interconversion of H_2_S and H_2_S_n_ due to redox reactions in vivo should be expected, and the single dose used by us may be more suitable to differentiate between the effects of these reactive sulfur species.

In conclusion, we demonstrated that not only H_2_S but also polysulfides are produced by the adipose tissue. Both reactive sulfur species are produced by different enzymes, have various biological activities and are differentially regulated by obesity. In particular, H_2_S stimulates lipolysis in a cAMP–PKA-dependent manner. Increased H_2_S production in the adipose tissue can contribute to stimulation of lipolysis and, in the long run, may accelerate fatty acid-mediated complications such as insulin resistance.

## 4. Materials and Methods

### 4.1. Animals and Reagents

All experiments were performed in 84 adult (2- to 2.5-month-old) male Wistar rats weighing 200–230 g. The study was approved by the Local Ethical Committee in Lublin (approval number 19/2019). Animals were kept at a temperature of 20 ± 2 °C under a 12-h light–dark cycle and had free access to standard laboratory chow and tap water. Animals were anesthetized by ethylurethane (1.25 g/kg ip) A thin polyethylene cannula (World Precision Instruments, Sarasota, FL, USA) was inserted into the carotid vein for continuous infusion of physiological saline (2 mL/hour) to avoid hypovolemia. The second cannula was inserted into the carotid artery for blood sampling. Their temperature was monitored by a rectal thermometer (World Precision Instruments, Sarasota, FL, USA) and was maintained at 36.5–37.5 °C using a heating table. Until otherwise stated, all reagents were obtained from Sigma-Aldrich (Steinheim, Germany).

### 4.2. Effect of H_2_S and Polysulfide Donors on Plasma NEFA, Glycerol, Glucose and Insulin

In the first set of experiments, we examined the effect of H_2_S donor, Na_2_S, and the polysulfide donor, Na_2_S_4_, on plasma NEFA, glycerol, glucose and insulin concentrations. After a 30-min stabilization period, the baseline blood sample was collected. Next, 0.5 mL of 0.9% NaCl (control), Na_2_S or Na_2_S_4_ was administered intravenously (at 100 μmol/kg in 0.5 mL 0.9% NaCl). Blood samples (0.5 mL) were collected after 15, 30, 45 and 60 min into tubes containing EDTA and were centrifuged at 2000× *g* for 5 min. Plasma was frozen and stored at −80 °C until the assay. Fifteen minutes after NaCl, Na_2_S or Na_2_S_4_ injection, slices of mesenteric adipose tissue were collected for the measurement of cyclic nucleotides and phosphodiesterase activity.

In additional animals, a dose-response effect of Na_2_S on NEFA and glycerol was studied after injecting various doses (1, 5, 10, 50, 100 or 200 μmol/kg) of this donor. Blood samples were collected 15 min after injection.

### 4.3. Effect of Cyclic Nucleotide-Dependent Protein Kinase Inhibitors on the Lipolytic Response to Na_2_S

The experiment was performed in 6 groups of rats receiving 0.5 mL 0.9% NaCl, the protein kinase A inhibitor KT5720 (1 μmol/kg) or the protein kinase G inhibitor KT5823 (1 μmol/kg) intraperitoneally and then, after 15 min, 0.5 mL 0.9% NaCl or 100 μmol/kg Na_2_S intravenously. Blood samples for the measurement of NEFA and glycerol were collected 15 min after the second injection.

### 4.4. Role of the Adrenergic System in the Regulation of Lipolysis by Na_2_S

The experiments were performed in 4 groups of rats receiving: (1) 0.5 mL 0.9% NaCl ip. and then, after 15 min, the β-adrenergic antagonist propranolol (1 mg/kg iv), (2) 0.5 mL 0.9% NaCl ip and then, after 15 min, the β-adrenergic agonist isoproterenol (1 mg/kg iv), (3) propranolol (1 mg/kg iv) and, after 15 min, isoproterenol (1 mg/kg iv), or (4) propranolol (1 mg/kg iv) and, after 15 min, Na_2_S (100 μmol/kg iv.). Blood samples for the measurement of NEFA and glycerol were obtained 15 min after the second injection.

### 4.5. Effect of Obesity on H_2_S and Polysulfide Production by Adipose Tissue

The study was performed in two groups of rats. The control group received standard chow (Agropol, Motycz, Poland providing 68% calories from carbohydrates 20% protein and 12% fat, whereas the second group received a high-calorie diet containing standard chow and a mixture of milk powder, sucrose, glucose and soybean powder (1:1:1:1) [42]. The composition of this diet (percentage of calories derived from carbohydrates, proteins and fat) is similar to that of normal chow, but the diet is highly palatable and increases food intake. The animals were fed with both diets for 1 month ad libitum. According to our previous studies [43], this protocol results in the development of obesity but there are still no significant metabolic abnormalities in animals fed the high-calorie diet. Animal were anesthetized, the abdominal cavity was opened, and blood samples were collected from the abdominal aorta for the measurement of insulin, glucose, lipids, NEFA and glycerol. Slices of mesenteric adipose tissue were collected for the measurement of H_2_S and H_2_S_n_ production.

### 4.6. Measurement of H_2_S and Polysulfide Production by Adipose Tissue

The generation of H_2_S by adipose tissue explants was measured by a H_2_S-selective sensor (ISO-H2S-2; World Precision Instruments, Sarasota, FL, USA) connected to a Four-Channel Free Radical Analyzer (TBR-4100) and a Lab-Trax-4/24T data acquisition system (World Precision Instruments) according to the manufacturer’s instructions. Each sensor was first calibrated by putting in the incubation solution without tissue and recording the current after adding increasing concentrations of Na_2_S. The calibration curve was constructed to calculate the sensor’s sensitivity (increase in current per unit of sulfide concentration).

After washing, slices of mesenteric adipose tissue (about 10 mg) were placed in 2-mL vials closed with tight caps containing a buffered Krebs–Ringer solution (114 mM NaCl, 5 mM KCl, 1.2 mM KH_2_PO_4_, 1.2 mM MgSO_4_, 1.0 mM CaCl_2_, 17 mM NaHCO_3_, 5.6 mM glucose and 16 mM Tris; pH 7.4) saturated with a mixture of N_2_ (90%), O_2_ (5%) and CO_2_ (5%). According to previous studies, H_2_S production is close to physiological at this O_2_ concentration [44]. The H_2_S sensor and a thin needle (for application of tested substances) were mounted through the caps so that their ends remained 5 mm above the vial bottom. After a 5-min equilibration period, substrates and/or inhibitors of H_2_S-synthesizing enzymes were added, and the measurement was started. The increase in current over time between 1 and 5 min after the addition of H_2_S donor was monitored, and the regression line (current = *a* × time + *b*) was fitted to the data by the least-squares method. H_2_S production was calculated from the slope (“*a*”) according to the sensitivity of the respective sensor. Each measurement was performed in two replications; in one of them, 1 mM dithiotreitol (DTT) was added to the incubation medium to reduce polysulfides to H_2_S. The signal measured in the sample without DTT was considered to be H_2_S, whereas that measured in the sample with DTT was considered to be total H_2_S+polysulfides. The amount of polysulfides produced was calculated as the difference between the samples containing and not containing DTT.

### 4.7. Measurement of NEFA Concentrations

Plasma non-esterified fatty acids were measured by the spectrofluorometric method using the kit provided by Cayman Chemical, Ann Arbor, MI, USA. In this method, fatty acids were first converted to acyl-CoA in the presence of Coenzyme A, ATP and acyl-CoA synthetase. Next, acyl-CoA is oxidized by acyl-CoA oxidase to 2,3-trans-enoilo-CoA and H_2_O_2_. Then, H_2_O_2_ reacted with 10-acetyl-3,7-dihydroxyphenoxazine in the presence of horseradish peroxidase to form resorufin, which was measured at the excitation/emission wavelengths of 535/590 nm. Fluorescence was measured with a microplate reader (PHERAstar FS, BMG Labtech, Orttenberg, Germany). Before the assay, plasma samples were diluted 3-fold. The NEFA concentration was calculated from the standard curve (0–0.25 mM oleic acid). The sensitivity of the method was 12 μM, whereas the intra- and inter-assay coefficients of variation were 2.6% and 3.9%, respectively.

### 4.8. Measurement of Glycerol Concentration

Plasma glycerol was measured spectrophotometrically using the Cayman Chemical kit (Ann Arbor, MI, USA cat. #10010755). Glycerol was first phosphorylated by glycerol kinase to glycerol-3-phosphate, which was then oxidized by glycerol-3-phosphate oxidase to dihydroxyacetone phosphate and H_2_O_2_. H_2_O_2_ then reacted with aminoantipyrine and N-ethyl-N-(3-sulfopropyl)-m-anisidine, forming a chromogenic quinoneimine compound, the absorbance of which was measured at 540 nm. The sensitivity of the method, and the intra- and inter-assay coefficients of variation were 13 μM, 4.5% and 6.8%, respectively.

### 4.9. Measurement of Cyclic Nucleotides in the Adipose Tissue

Adipose tissue slices were homogenized in 50 mM NaCl buffered with a phosphate buffer (pH 7.4) containing 10 μM of the phosphodiesterase inhibitor 3-isobutyl-1-methylxanthine (IBMX) to inhibit cAMP and cGMP hydrolysis during sample processing (100 μL of buffer/10 mg tissue). The homogenate was centrifuged at 14 000× *g* for 10 min at 4 °C and diluted 100-fold. Cyclic AMP and cyclic GMP concentrations were measured immunoenzymatically by Cayman Chemical kits (Ann Arbor, MI, USA cat.# 581001 and 581121, respectively). The sensitivity, intra-assay CV and inter-assay CV values for cAMP and cGMP were 0.1 pmol/mL, 5.1% and 7.0%, and 0.9 pmol/mL, 4.7% and 7.7%, respectively.

### 4.10. Measurement of Phosphodiesterase (PDE) Activity in the Adipose Tissue

cAMP-hydrolyzing PDE activity was measured using a kit purchased from BioVision (Milpitas, CA, USA, catalogue number K-2013-10). Ten mg of adipose tissue was homogenized in 100 μL of a buffer containing in the kit and centrifuged at 1000× *g* for 15 min at 4 °C. The supernatant (20 μL) was transferred to microplate wells and was diluted with 30 μL of a phosphate buffer. Each sample was measured in 2 replications; one of them contained 2 μL of a cAMP solution. The product (AMP) was converted to a fluorescent compound by the mixture of enzymes and reagents provided with the kit. Fluorescence was measured at the excitation and emission wavelengths of 535 nm and 587 nm, respectively, in the kinetic mode for 30 min. AMP concentration was calculated from the standard curve (0–1000 pmol AMP/well). The increase in fluorescence between 5 and 15 min in the sample that did not contain cAMP was subtracted from the increase in fluorescence in the sample containing cAMP. Enzyme activity was expressed in pmol AMP/min per mg protein.

### 4.11. Measurement of Glucose, Lipids and Insulin

Plasma glucose, triglycerides and total cholesterol were measured with kits purchased from Alfa Diagnostics (Warsaw, Poland). Plasma insulin was measured immunoenzymatically using the Mercodia kit (cat. #10-1250-01). The sensitivity, intra-assay and inter-assay CV values for insulin measurement were 0.15 μg/L, 3.1% and 4.4%, respectively. The anti-insulin antibodies contained in the kit exhibited 7% cross-reactivity with rat proinsulin and 0.001% cross-reactivity with rat C peptide.

### 4.12. Statistical Analysis

The results are expressed as the means ± SD of 6 animals/adipose tissue samples per group. Between-group comparisons were performed by Student’s t-test or ANOVA, followed by Tukey’s test for 2 groups and >2 groups, respectively. The results from the same group obtained at different time points were analyzed by ANOVA for related variables. The area under the curve (AUC) of plasma NEFA and glycerol concentrations was calculated by the trapezoidal method, taking the baseline concentration in the control group as 0. A *p* < 0.05 was considered statistically significant.

## Figures and Tables

**Figure 1 ijms-23-01346-f001:**
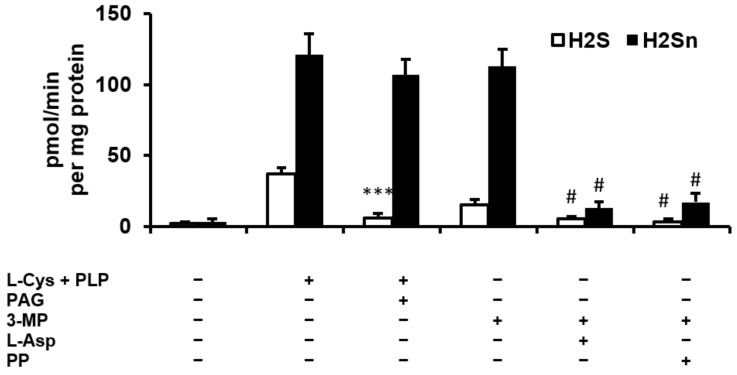
Hydrogen sulfide (H_2_S) and polysulfide (H_2_S_n_) production by adipose tissue slices ex vivo in the presence or absence of L-cysteine (L-Cys, 1 mM), pyridoxal 5′-phosphate (PLP, 1 mM), propargylglycine (PAG, 1 mM), 3-mercaptopyruvate (3-MP, 5 mM), L-aspartate (L-Asp, 3 mM) and phenylpyruvate (PP, 10 mM). *** *p* < 0.001 vs. sample incubated in the presence of L-Cys + PLP, ^#^
*p* < 0.001 vs. sample incubated in the presence of 3-MP.

**Figure 2 ijms-23-01346-f002:**
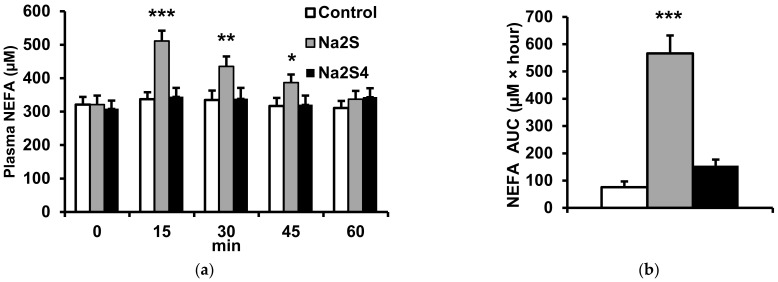
Effect of Na_2_S and Na_2_S_4_ on plasma non-esterified fatty acids (NEFA). Plasma NEFA concentrations at different time points after injection of 0.9% NaCl (control), Na_2_S (100 μmol/kg) or Na_2_S_4_ (100 μmol/kg) (**a**). The area under the curve (AUC) of plasma NEFA concentration was calculated for 0–60 min period after NaC, Na_2_S or Na_2_S_4_ administration (**b**). * *p* < 0.05, ** *p* < 0.01, *** *p* < 0.001 vs. baseline level in the control group.

**Figure 3 ijms-23-01346-f003:**
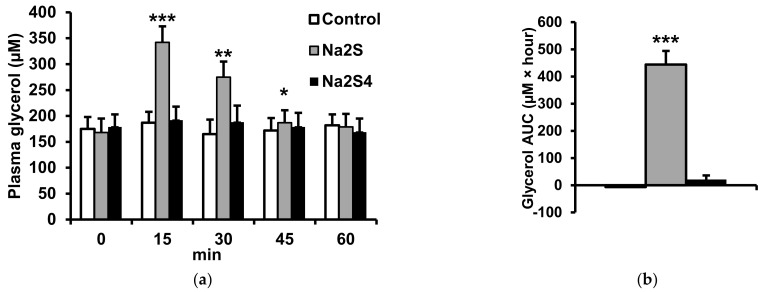
Effect of Na_2_S and Na_2_S_4_ on plasma glycerol. Plasma glycerol concentrations were measured at different time points after the injection of 0.9% NaCl (control), Na_2_S (100 μmol/kg) or Na_2_S_4_ (100 μmol/kg) (**a**). The area under the curve (AUC) of plasma glycerol concentration was calculated for a 0–60 min period after NaC, Na_2_S or Na_2_S_4_ administration (**b**). * *p* < 0.05, ** *p* < 0.01, *** *p* < 0.001 vs baseline level in the control group.

**Figure 4 ijms-23-01346-f004:**
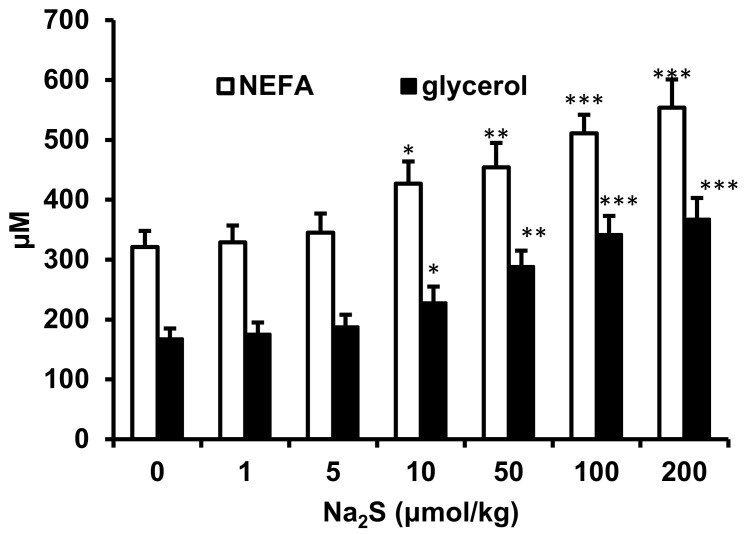
Dose-dependent effect of Na_2_S on plasma NEFA and glycerol concentrations measured 15 min after injection. * *p* < 0.05, ** *p* < 0.01 *** *p* < 0.001 vs. values before Na_2_S administration.

**Figure 5 ijms-23-01346-f005:**
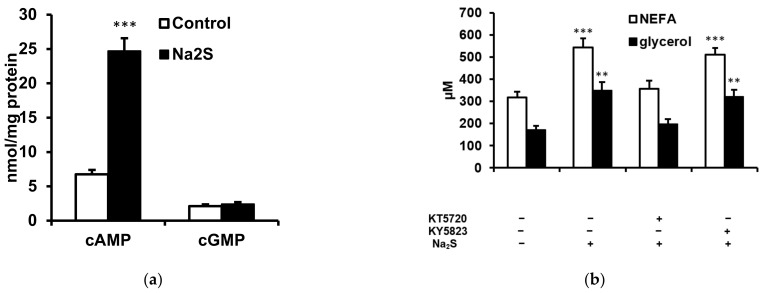
Role of cyclic nucleotides and cyclic nucleotide-dependent protein kinases in the regulation of lipolysis. (**a**) Effect of Na_2_S (100 μmol/kg) on cAMP and cGMP concentrations in the adipose tissue. (**b**): Effect of the PKA and PKG inhibitors KT5720 and KT5823, respectively, (each administered at 1 μmol/kg) on NEFA and glycerol concentrations in animals injected with 100 μmol/kg Na_2_S. ** *p* < 0.01, *** *p* < 0.001 vs. rats not receiving Na_2_S.

**Figure 6 ijms-23-01346-f006:**
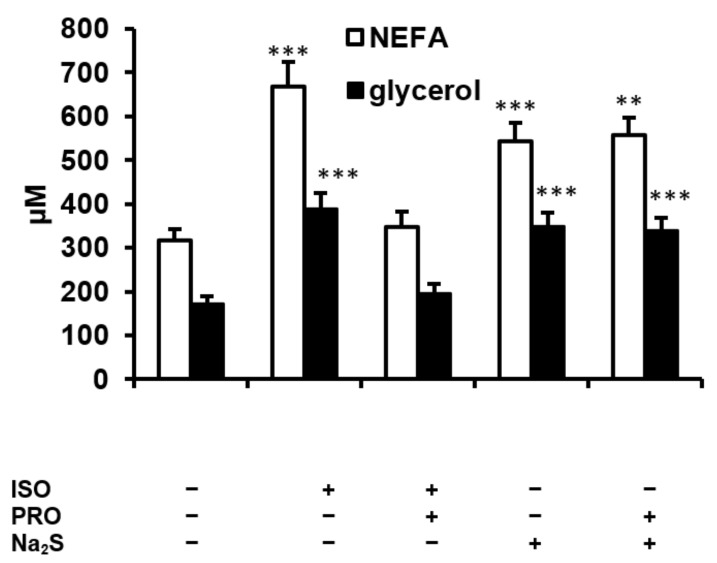
Effect of the β-adrenergic agonist isoproterenol (ISO) and the β-adrenergic antagonist propranolol (PRO) on plasma NEFA and glycerol concentrations in rats treated with or without Na_2_S (100 μmol/kg). ** *p* < 0.01, *** *p* < 0.001 vs. control group not receiving Na_2_S, isoproterenol or propranolol.

**Table 1 ijms-23-01346-t001:** Effect of obesity on metabolic parameters, H_2_S and polysulfide production in adipose tissue.

	Control	Obese
Body weight (g)	221 ± 16	268 ± 18 **
Triglycerides (mM)	0.81 ± 0.05	0.88 ± 0.08
Total cholesterol (mM)	2.08 ± 0.23	2.15 ± 0.27
NEFA (μM)	352 ± 25	714 ± 46 ***
Glycerol (μM)	172 ± 15	252 ± 19 ***
Insulin (μU/mL)	21.3 ± 2.5	23.4 ± 2.9
Glucose (mM)	4.22 ± 0.51	4.44 ± 0.58
H_2_S production (pmol/min per mg protein)	32.7 ± 3.5	72.6 ± 6.1 ***
Polysulfide production (pmol/min per mg protein)	116.4 ± 26.7	101.7 ± 19.4

** *p* < 0.01, *** *p* < 0.001 vs. control group.

## Data Availability

Original data are available from the corresponding author on request.

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
