# Peer review of "Role of Hydrogen Sulfide and Polysulfides in the Regulation of Lipolysis in the Adipose Tissue: Possible Implications for the Pathogenesis of Metabolic Syndrome"

_ijms, 2022, doi:10.3390/ijms23031346_

Round 1
Reviewer 1 Report
The topic of this manuscript falls within the scope of International Journal of Molecular Sciences.
The role of H2S in the regulation of adipose tissue lipolysis has been examined in few studies so far. Therefore, the topic of the presented manuscript is very important, relevant, and original. The study is very important for scientific community.
The aim of the paper was to examine the effect of H2S and polysulfides on adipose tissue lipolysis.
On the basis of obtained results the Authors showed that: both H2S and polysulfides are produced in the adipose tissue and under physiological conditions the amount of polysulfides is higher than H2S, andH2S stimulates lipolysis in cAMP-PKA dependent but β-adrenergic as well as obesity induced by high-calorie diet increases H2S but not H2S production in the adipose tissue.
The data has been provided with vigorous statistical analysis. The Authors have presented sufficient data. The appropriate tables and figures have been provided. The manuscript is well written. The article is easy to read and logically structured. The methods are adequately described. The Authors also added limitations in the discussion section. The conclusions are consistent with presented evidence and arguments. They address the main question posed.
In my opinion, this is a quite interesting paper, which could be published after addressing by the research group a few minor comments specified below:
- In the introduction, please mention also about perivascular adipose tissue, which play very important role in vascular dysfunction [the newest papers in this field: The Role of Obesity-Induced Perivascular Adipose Tissue (PVAT) Dysfunction in Vascular Homeostasis. Nutrients. 2021 Oct 28;13(11):3843. doi: 10.3390/nu13113843.; Roles of Perivascular Adipose Tissue in Hypertension and Atherosclerosis. Antioxid Redox Signal. 2021 Mar 20;34(9):736-749. doi: 10.1089/ars.2020.8103.]
2. Please also add the newest following papers:
Energy metabolism in brown adipose tissue. FEBS J. 2021 Jun;288(12):3647-3662. doi: 10.1111/febs.16015
Human Brown Adipose Tissue and Metabolic Health: Potential for Therapeutic Avenues. Cells 2021, 10, 3030. https://doi.org/10.3390/cells10113030
Author Response
Dear Reviewer,
Thank you very much for evaluating our manuscript and for your positive opinion. Your comments were very helpful for us to revise the paper. Below are listed all the corrections made in the manuscript as well as our responses to your comments.
Comment 1
In the introduction, please mention also about perivascular adipose tissue, which play very important role in vascular dysfunction [the newest papers in this field: The Role of Obesity-Induced Perivascular Adipose Tissue (PVAT) Dysfunction in Vascular Homeostasis. Nutrients. 2021 Oct 28;13(11):3843. doi: 10.3390/nu13113843.; Roles of Perivascular Adipose Tissue in Hypertension and Atherosclerosis. Antioxid Redox Signal. 2021 Mar 20;34(9):736-749. doi: 10.1089/ars.2020.8103.].
Response
We have mentioned the role of PVAT in the Introduction by adding the following text:
“Specific adipose tissue depots have also special functions. For example, perivascular adipose tissue which surrounds blood vessels produces vasodilating and antiinflammatory mediators which have an important role in maintaining vascular homeostasis, however, obesity is associated with PVAT dysfunction due to its inflammation and local oxidative stress”. The papers suggested by the Reviewer are cited and added to the reference list as ref. [5] and [6].
Comment 2.
Please also add the newest following papers:
Energy metabolism in brown adipose tissue. FEBS J. 2021 Jun;288(12):3647-3662. doi: 10.1111/febs.16015
Human Brown Adipose Tissue and Metabolic Health: Potential for Therapeutic Avenues. Cells 2021, 10, 3030. https://doi.org/10.3390/cells10113030.
Response
We have mentioned the role of BAT in the Introduction by adding the following sentence:
“Unlike white adipose tissue, brown adipose tissue (BAT) contains many small lipid droplets and a lot of mitochondria and is characterized by intensive fatty acid oxidation through the uncoupled mitochondrial respiration with little ATP production and a lot energy dissipated as heat making it important in thermogenesis. Interestingly, in addition to fatty acids BAT can oxidize other metabolites such as glucose, lactate, succinate and branch-chain aminoacids [BAT1]. BAT also secretes mediators which improve metabolism of remote tissues referred to as BATokines”. The papers suggested by the Reviewer are cited and added to the reference list as ref. [7] and [8].
In addition, reference numbering has been updated throughout the text and the reference list.
We hope you will find the revised manuscript improved and suitable for publication.
Sincerely
Jerzy Beltowski
Reviewer 2 Report
BeÅ‚towski and Wiórkowski showed the role of H2S in the regulation of lypolisis in adipose tissue. The experiments are appropriate, the pharmacological approach dissect the mechanisms underlying lypolisis, but there are some concerns to be addressed:
- The title does not reflect the true scope of this work. In my opinion this is a carefully conducted study, however the metabolic syndrome is not a target of paper. For metabolic syndrome some parameters and tissue analysis must be performed. This study is limited to adipose tissue.
- H2S donors. Slow releasing H2S donor GYY4137 has been extensively studied in cardiovascular field (doi: 3390/antiox10030486; doi.org/10.3389/fphar.2021.613989). A slow-releasing H2S GYY4137 is considered more physiologically H2S donor.
Author Response
Dear Reviewer,
Thank you very much for evaluating our manuscript and for your positive opinion. Your comments were very helpful for us to revise the paper. Below are listed all the corrections made in the manuscript as well as our responses to your comments.
Comment 1
The title does not reflect the true scope of this work. In my opinion this is a carefully conducted study, however the metabolic syndrome is not a target of paper. For metabolic syndrome some parameters and tissue analysis must be performed. This study is limited to adipose tissue.
Response
We remained the original title with only a minor change:
“Role of hydrogen sulfide and polysulfides in the regulation of lipolysis in the adipose tissue: Possible implications for the pathogenesis of the metabolic syndrome”.
The title clearly indicates that we examined the effect of H2S and polysulfides on lipolysis in the adipose tissue. The second part of the title was inspired by the results which show that: (i) H2S stimulates lipolysis, (ii) the amount of H2S is higher in obese rats. These results suggest that H2S may be involved in augmented lipolysis in obesity which contributes to many components of the metabolic syndrome mediated by the excess of non-esterified fatty acids. The first part of the title refers strictly to the results whereas the second part to these implications. We added the word “possible” to indicate that these implications are only suggested at this stage. We believe that the title well represents the study.
Comment 2.
H2S donors. Slow releasing H2S donor GYY4137 has been extensively studied in cardiovascular field (doi: 3390/antiox10030486; doi.org/10.3389/fphar.2021.613989). A slow-releasing H2S GYY4137 is considered more physiologically H2S donor.
Response
We cite these two studies in the revised Discussion (ref. 29 and 30. We agree that GYY4137 is commonly used as the slowly-releasing H2S donor. However, we used Na2S for the following reasons: (i) GYY4137 may have H2S-independent effects mediated by its parent molecule, (ii) at least one study has demonstrated that GYY4137 does not release H2S effectively in the rat plasma, (iii) Na2S and Na2S4 are closed structurally whereas there is no polysulfide-releasing GYY4137 analogue, therefore, using inorganic salts seems more reasonable when the effects of both reactive sulfur species are to be compared. This explanation is now added to the Discussion.
In addition, reference numbering has been updated throughout the text and the reference list.
We hope you will find the revised manuscript improved and suitable for publication.
Sincerely
Jerzy Beltowski
Reviewer 3 Report
The study conducted by Jerzy BeÅ‚towski and Krzysztof Wiórkowski demonstrated that H2S stimulates lipolysis in cAMP-PKA dependent manner. However the evidences on the role of H2S in the adipose tissue are controversial; for these reason I have many cquestions:
- H2S production increase with rat age so can you specify the age of the animals? Have you verified these results in young rat?
- Have you the possibility to demonstrate an increase in mRNA levels of cystathionine β-synthase (CBS), cystathionine gamma-lyase (CSE) and cysteine aminotransferase together with 3-mercaptopyruvate sulfurtransferase (3-MST) in adipose depot
- In your opinion is there a possible correlation between autophagy and H2S level ?
- Concerning inflammation/oxidative stress markers is there a relationship with H2S level?
- Please in introduction section: line 32 I suggest this recent pubblication concerning the endocrine role of adipose tissue PMID: 33255520; line 38 concerning the effects of metabolic syndrome I suggest also kidney damage described in this paper PMID: 30884780
Author Response
Dear Reviewer,
Thank you very much for evaluating our manuscript and for your positive opinion. Your comments were very helpful for us to revise the paper. Below are listed all the corrections made in the manuscript as well as our responses to your comments.
Comment 1
H2S production increase with rat age so can you specify the age of the animals? Have you verified these results in young rat?
Response
The study was performed in rats aged 2-2.5 months. This information was added in the Methods (section 4.1). The effect of age on H2S production in the adipose tissue is controversial. While some studies found increase in H2S with age, Katsouda et al. (Pharmacological Research 2018; 128:190-199) have demonstrated that H2S concentration in the subcutaneous and visceral (epidydimal) white adipose tissue tended to be lower in 24 week-old than in 8-week old mice. It would be reasonable to examine the effect of age on the regulation of lipolysis by reactive sulfur species, however, it was not a focus of the present study.
Comment 2.
Have you the possibility to demonstrate an increase in mRNA levels of cystathionine β-synthase (CBS), cystathionine gamma-lyase (CSE) and cysteine aminotransferase together with 3-mercaptopyruvate sulfurtransferase (3-MST) in adipose depot?
Response
In the ongoing research we measure the expression of these enzymes in the adipose tissue. Preliminary results suggest that their expression does not change in this model of obesity. These results are not presented in this study. The aim of the ongoing study is to examine the mechanism of H2S increase in obese rats and we measure a lot of other things to elucidate this mechanism including CBS, CSE and 3-MST activities, mitochondrial H2S oxidation, the expression of enzymes involved in H2S oxidation, coenzyme Q (the cofactor of H2S oxidation), markers of mitochondrial density and biogenesis, etc. All these results will be presented together in the separate manuscript.
Comment 3
In your opinion is there a possible correlation between autophagy and H2S level ?
Response
Thank you very much for this comment. This is a very important issue. Apart from canonical lipolysis accounted for by TGL, HSL and MTGL mentioned in the Introduction, triglycerides contained in lipid droplets may be hydrolyzed by autophagy (referred to as lipophagy). There are three types of lipophagy: macro-, micro- and chaperone-mediated lipophagy. Macrolipophagy is initiated by engulfment of lipid droplet fragments in lysosomes followed by their hydrolysis by lysosomal acid lipase. In contrast, micro- and chaperone-mediated lipophagy are not associated with lipid engulfment but are dependent on the interaction of lipid droplets with lysosomes and TGL may be involved in both of them.
Autophagy is stimulated in the adipose tissue of obese animals and humans, however, this was mostly concluded according to the expression of autophagy-related proteins. Stimulation of autophagy results in the stimulation of lipolysis in adipocytes, therefore, it cannot be excluded that increased lipolysis in obese rats results from stimulation of both lipolysis and lipophagy. In addition, because PKA may activate ATGL by phosphorylating perilipin-5, increase in intracellular cAMP could stimulate at least microlipohagy and chaperone-mediated lipophagy. The effect of H2S on autophagy is controversial; both stimulatory and inhibitory effects have been described in various systems. At present it is unclear if H2S has any role in the regulation of lipophagy and what is the quantitative contribution of canonical lipolysis and lipophagy to fatty acid and glycerol release in lean and obese rats. This could be an interesting area for future research but this issue was not addressed experimentally by us in the present study and therefore is not mentioned in the Discussion.
Comment 4
Concerning inflammation/oxidative stress markers is there a relationship with H2S level?
Response
This is a very important and complex issue. Adipose tissue of obese animals and humans is characterized by inflammatory reaction and local oxidative stress, and inflammatory mediators may contribute to enhanced lipolysis by increasing the expression of lipases [Nat Metab. 2021 Nov;3(11):1445-1465]. Most studies suggest the anti-inflammatory effect of H2S, however, its proinflammatory effects have also been reported. In the present study we did not address the role of inflammatory reaction and oxidative stress in the effect of H2S. However, it is unlikely that pro-inflammatory effect of H2S could contribute to stimulation of lipolysis because fatty acids and glycerol were measured soon after administration of Na2S and, as stated above, inflammatory cytokines up-regulate lipolytic enzymes at the transcriptional level which requires more time to occur. In the ongoing study we are addressing the effect of H2S and polysulfides on adipose tissue inflammation. Interestingly, the results suggest opposite pro- and anti-inflammatory effects of these sulfur species, respectively, however, the study is not finished yet and the results will be presented in the separate manuscript. Regarding oxidative stress, H2S is strong reducing agent and was suggested to serve as the endogenous antioxidant, however, due to its very low concentrations in tissues it is unlikely that this effect is functionally very relevant.
Comment 5
Please in introduction section: line 32 I suggest this recent publication concerning the endocrine role of adipose tissue PMID: 33255520; line 38 concerning the effects of metabolic syndrome I suggest also kidney damage described in this paper PMID: 30884780
Responses
These papers were added to the reference list [4, 10] and are cited in the appropriate places in the Introduction. Nephropathy is also mentioned among obesity complications.
In addition, reference numbering has been updated throughout the text and reference list.
We hope you will find the revised manuscript improved and suitable for publication.
Sincerely
Jerzy Beltowski
Round 2
Reviewer 2 Report
None
Author Response
There are no critical comments from the Reviewer 2.